# Some Hard or Soft Coatings to Protect the Pristine Biometallic Substrates under Fretting-Corrosion Solicitations: What Should Be the Best Solution?

**Jean Geringer [1],\*, Vincent Fridrici [2] , Haohao Ding [2], Kyungmok Kim [3] , T. Taylor [4], Lerato Semetse [4], Sara Ehsani-Majd [1], Peter Olubambi [4], Julien Fontaine [2] and Philippe Kapsa [2]**

[1]   Mines Saint-Etienne, Center for Health and Engineering, Inserm U1059, 158 cours Fauriel, 42023 Saint-Etienne, France; sara.ehsani-majd@emse.fr

[2]   Ecole Centrale de Lyon, LTDS – UMR CNRS 5513, 36 avenue Guy de Collongue, 69134 Ecully, France; vincent.fridrici@ec-lyon.fr (V.F.); dinghaohao520@163.com (H.D.); julien.fontaine@ec-lyon.fr (J.F.); philippe.kapsa@ec-lyon.fr (P.K.)

[3]   School of Aerospace and Mechanical Engineering, Korea Aerospace University, 100 Hanggongdae gil, Hwajeon-dong 412-791, Korea; kkim@kau.ac.kr

[4]   Faculty of Engineering and the Built Environment, 4140, John Orr Building PO Box 524, Auckland Park 2006, South Africa; tahlitatayler14@gmail.com (T.T.); lerato.semetse@gmail.com (L.S.); polubambi@uj.ac.za (P.O.)

\*   Correspondence: geringer@emse.fr

**Abstract:** Under tribological conditions in aqueous medium, the contact of materials does involve some degradations of materials. Especially friction under small reciprocal displacement, i.e., fretting corrosion, is occurring; this topic has been highlighted since the 80′s regarding hip implants. Hip prosthesis is assembled from three parts: femoral stem, neck and head. Fretting corrosion or friction corrosion between metallic parts first involves some degradation of the oxides layers. This step is governed by mechanics and it is related to some few minutes. Afterwards the corrosion occurrs enhanced by mechanical degradation. As well focused some oxides and some metallic ions are related to biocompatibility issues. Some strategies are available in order to avoid metal against metal friction and/or fretting. Some hard coatings and some smooth coatings were investigated. The first one is diamond-like carbon (DLC), and the second is a polyetheretherketone (PEEK), polymeric one. The investigations were focused on fretting corrosion solicitations of Ti-6Al-4V vs. Ti-6Al-4V + coating. DLC as a coating delays the corrosion degradation. The PEEK coating does not promote any corrosion degradation of the metallic counter part and more generally any wear.

**Keywords:** tribocorrosion; coatings; metallic biomaterials; PEEK; DLC

## 1. Introduction

Hip joint, Figure 1, is one of the most practiced surgical operations, i.e., 160,000 total hip arthroplasty (THA) operations in France (order of magnitude coming from [1]). All over the world, there are more like than 1.5 million every year.

Every patient is different from another one due to morphology, size, body mass index, etc. Due to manufacturing issues, getting one implant dedicated to one patient sounds unrealistic concerning economical issues. Thus, the modularity did appear as joining both issues: personalized implants and reducing manufacturing costs. By getting one stem, one neck and every hip component related to every patient anatomy does appear to be the sustainable way. Figure 2 shows some examples of some modularity. In the USA, the trunnion vocable is well evoked concerning this issue.

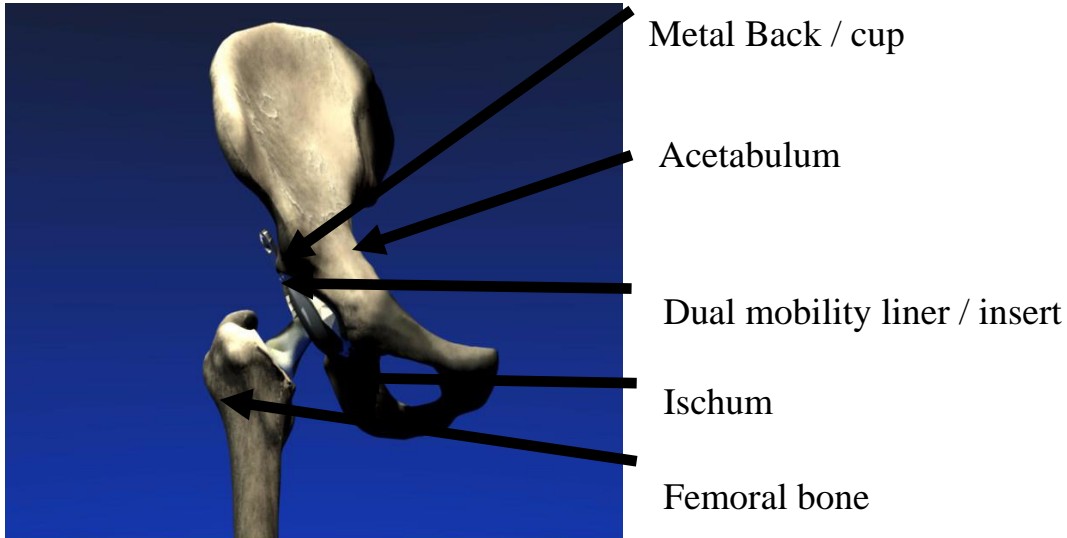

**Figure 1.** Total hip joint, schematic view, dual mobility.

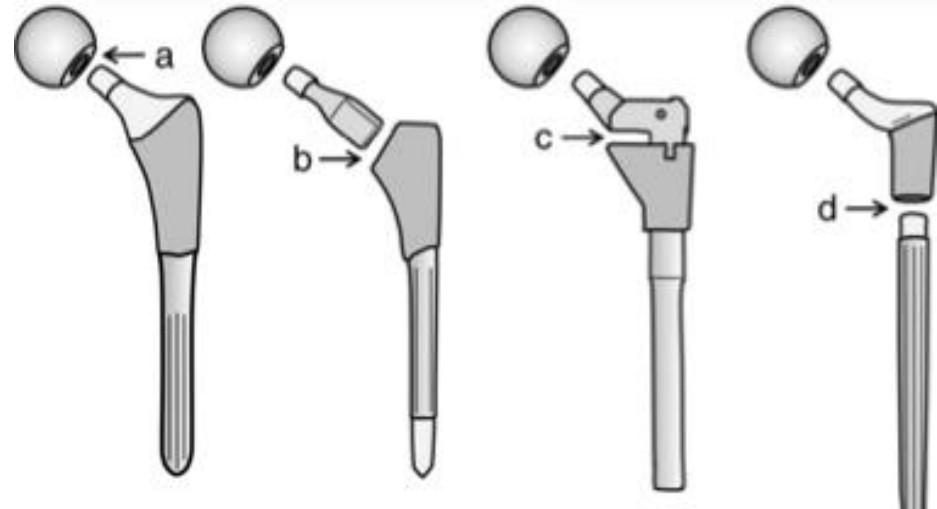

**Figure 2.** (**a**) Usual femoral component as hip joint; (**b**) modular hip joint as adapting the size; (**c**) modular hip joint as adapting the angle of the femoral neck; (**d**) modular hip joint as adapting the stem length [2].

As shown in Figure 2, one may notice the metallic junction that is occurring on every implant. During the human gait, some alternative loadings will be imposed on every metallic junction. By having a look at the ISO 14242 standard and the ASTM F1714 [3,4], the load amplitude should reach from 300N to 5000N. Every metallic junction does not match disassembling, except the positioning issue; however, some small relative displacements will occur on every metallic junction, i.e., few dozens of microns. Moreover, total hip joints are immersed among physiological liquid protein solutions. First, one may imagine some metals involved in a metallic junction. The mechanical properties are quite different, 115 GPa concerning the Ti-6Al-4V and 200 GPa concerning the 316L stainless steel or the Co-Cr-Mo, between materials in contact. Under applied stress, the deformation should be different and the friction/fretting (friction under small displacements) should take place. Second, even if the material is quite the same, Ti-6Al-4V concerning femoral stem, the junction roughness, Sa value, will vary from 1650 nm to 2250 nm ± 30 nm [5]. On some surface peaks (small contact area), the contact stress may also reach some GPa. Due to metal plasticity the mechanical properties on the top surface should be different from the ones related to the bulk material. Under applied stresses some micro relative displacements may also occur. This last phenomenon is not predominant through

the machining expertise of some manufacturers. However due to economical reasons the implants dealers are contracting with some sub-manufacters and the quality check might be sometimes not achieved. After the metal on metal issue between head and cup junction [6], some awkwardness in the manufacturing process did happen. In order to face this issue, some researchers took into account the coating impact on one material involved in the metallic junction.

## 2. Coatings Statements

Two coatings will be investigated through this work: diamond-like carbon (DLC) and polyetheretherketone (PEEK). DLC is well used in the automotive industry. Through this chapter, no process details are going to be presented relating to IREIS[TM] property, or even DLC and PEEK. The authors will instead present some general comments about both coatings.

DLC, diamond-like carbon, is one of the famous materials advances in order to decrease the friction coefficient, during friction phenomenon [7]. It should be compared as ice on the metallic surface but DLC is an amorphous structure in contrast to ice. Carbon atom should be hybridized $sp^2$ and or $sp^3$ and the hydrogen quantity is adapted according to the hybridization state [8,9]. When some metal is involved in a lubricated contact, especially with water, it is known that metal corrosion should play one of the major roles in the mechanical degradation of the metallic wear. Avoiding destruction of the oxides layer on the top surface of metal is possible thanks to DLC coating. Through the 80s and 90s, industry focused on improving hardness and mechanical properties of any contact, DLC coating was focused on improving the metallic and some ceramic surfaces [10–12]. As mentioned in [13], DLC is well managed in aerospace and space environment under severe mechanical conditions. One may suggest that the DLC coating should be the best candidate in order to improve some implants durability. Many investigations were practiced from the 2000's in the implants area in order to avoid some metallic degradations, involving some benefits. Some results [14–19] showed the expected benefits from DLC coatings dedicated to implants. Hip, knee and dental implants that are submitted to friction, i.e., some tribological issues, are concerned. Unfortunately, some catastrophic failures did appear. Concerning implants inside the human body, keeping in mind that their lifetime should be around 20 years, many patients suffered instead of taking some benefits from implants. That case was especially related to hip joint patients. Hauert et al. [20–22] surveyed some DLC statements and improvements concerning the implants field during a 10-year period. Thus, some improvements thanks to deep research investigations on DLC manufacturing were accomplished [23,24]. Notwithstanding that some attempts on finger joints were achieved but some doubts were also consistent about hip joints. The surgical community, especially orthopedic surgeons, does not reintroduce DLC in this implants area. Nowadays three methods are well investigated in order to deposit DLC coatings: Plasma-Activated Chemical Vapor Deposition (PACVD), Physical Vapor Deposition (PVD) by magnetron sputtering, and Physical Vapor Deposition by cathodic arc, unfiltered and filtered.

That is the reason why, even considering the low roughness of metallic materials that facilitate protection from wear/corrosion degradations, some coatings nowadays need to be improved. DLC is the hardest one; by changing the strategy, one may think about getting a smooth coating. The guidelines do contain the requirements related to biocompatibility, mechanical integrity, tribological performances and aging properties. Thus, a quite good candidate should be biocompatible polymeric material. With time and experience (some improvements through some wear rate behavior [25]) PEEK, PEEK the required candidate. The sputtering method is quite different than the one related to DLC. Due to a PEEK fusion temperature around 370 °C PACVD or PVD methods should be considered as destructive ones. Thus, the onsite method is quite close to a paint gun with some electrostatic effects in order to be sure that the PEEK pellets adhere on the PEEK surface. As with DLC, the process shall not be detailed through this work due to confidentiality agreements. However, some typical properties related to PEEK are relevant; one may notice a phenomenon close to the Schallamach waves [26,27], especially after some tribological solicitations. This issue should be related to some difficulties in order to measure some wear volume. That is the reason why, through this work, some wear volume results

will be presented but the main statement is that the surface state of PEEK should be changed according to wear solicitations and/or to these typical waves.

During this chapter, the authors will not pay attention to the manufacturing process of every coating. Notwithstanding the physical experimental tests in order to discriminate the coating efficiency will be described and they will be emphasized in order to avoid the impact of some reciprocal displacement between materials in contact.

## 3. Materials and Methods

The specified experimental test, as called fretting corrosion, will be emphasized through the strategy for improving the mechanical parameters and the electrochemical ones. The degradation is dual: a mechanical one required in order to destroy the oxides layer and an electrochemical one that takes into account the metallic dissolution. The materials should be described and especially the key factors to promote the expected coating efficiency.

### 3.1. Fretting Corrosion Device

The Figure 3 is showing the fretting corrosion machine (TA instruments Inc, New Castle, De, USA) and some elements around. The motor is electromagnetic, ex. Bose™ TAinstruments™. The displacement transducer capacity is maximum ±400 μm.

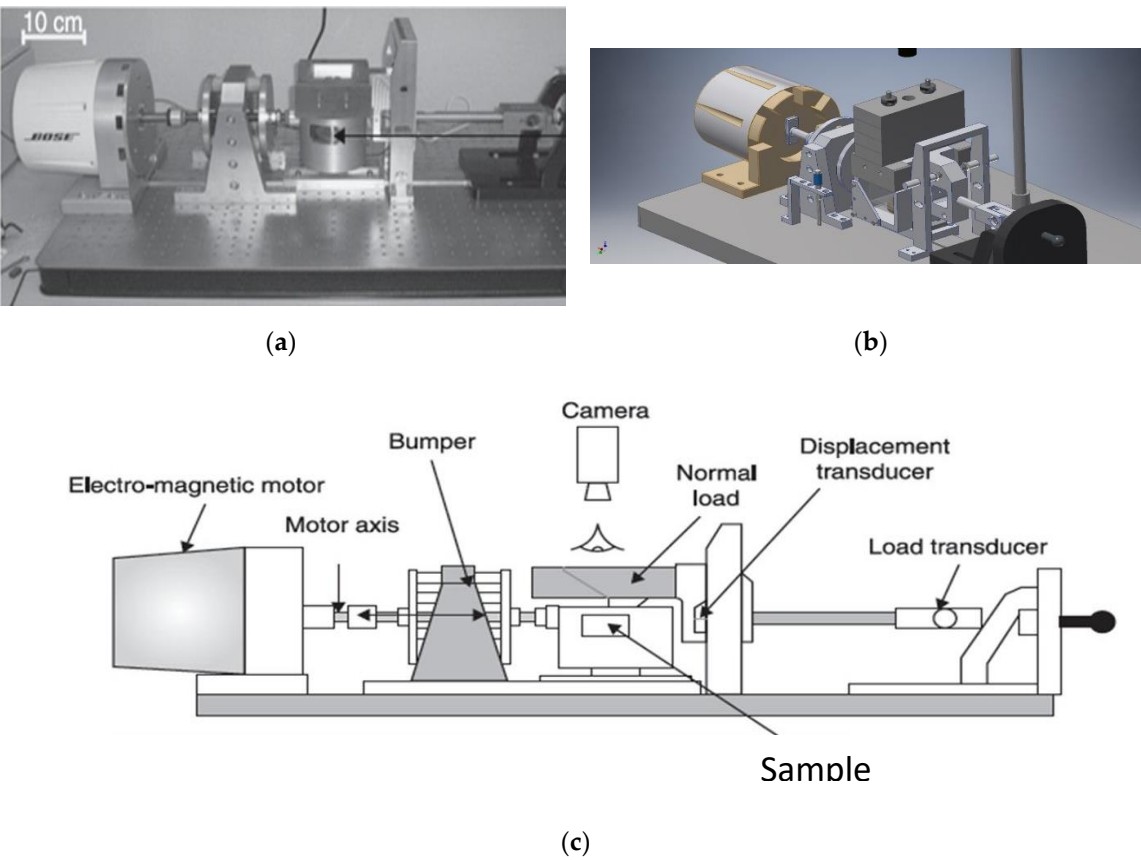

(**a**)        (**b**)

(**c**)

**Figure 3.** (**a**) The fretting corrosion machine; (**b**) schematic view, right side; (**c**) fretting corrosion machine with legends.

The Figure 3a,b are representing the global overview of the device. The Figure 3c is showing a scheme with legends. The bumper is a key element in order to accommodate the relative displacement during the friction process, i.e., few microns. The fixed sinusoidal displacement amplitude is ±40 μm during the whole campaign dedicated to this survey. The normal load is driven through the contact

pressure highlighted in modular junction. As an example, concerning the neck–stem connections, some modeling predicted until 600 MPa. The fretting corrosion machine is not manufactured in order to usually apply this kind of load. By bending through a friction coefficient between materials in contact of around 0.5, an overload into the machine control system evolved. Thus, in order to apply the realistic normal load/stresses, instead of changing the motor and/or some machine elements, the samples sizes were newly designed. The contact surface was decreased until reaching 750 MPa of contact stresses. A cylindrical shape was manufactured, obtaining 1 mm of cylindrical contact length. Moreover, the roughness of the manufactured samples was controlled by accurate polishing steps, in a homemade way. The drawback of some electromagnetic motors is related to the so low tangential load that is applied compared with hydraulic machines. An usual 10 kN machine capacity is required on any traction compression machine, such as an experimental one (table machine). The daily usage of this kind of machine is dedicated to study some biological tissues (muscles, tendons, etc.) Hopefully, due to max. 750 N of normal load that is required to mimic the actual contact, the Bose machine was well designed (machine and samples) and dedicated to reproduce a typical modular junction of hip prosthesis. From this state of the art concerning the machine, some investigated materials shall be detailed.

### 3.2. Materials

This paper pays attention to the actual modular junction. The authors were focusing on a Ti-6Al-4V/Ti-6Al-4V junction; typically it is considered from the femoral stem, case c and d on Figure 2. As mentioned in the preliminary part, two strategies of coatings were practiced. Table 1 highlights some basic mechanical performances for comparing Ti-6Al-4V and the investigated coatings.

**Table 1.** An overview of mechanical properties related to investigated materials.

| | Elastic Modulus/GPa | Poisson Ratio | Elastic Limit/MPa | Yield Strength/MPa | KIC/MPa·m$^{1/2}$ | Hardness |
|---|---|---|---|---|---|---|
| Ti-6Al-4V* | 115 | 0.34 | 880 | 950 | 90 | 36, rockwell C |
| DLC | 1000** | 0.22–0.33** | 2.2–1.1*** | 2.5*** | 10*** | 10–80** GPa |
| PEEK* | 3.6 | 0.39 | 100 | 172 | 1–2 | 126, rockwell R |

*: [28]; **: [29,30]; ***: [31,32].

Under fretting corrosion solicitations, the samples were immersed in the physiological liquid, i.e., a simulated one, bovine serum (Biowest$^{TM}$, regulation (EC) 999/2001). Thus, an electrochemical setup was used to perform Open Circuit Potential through the measurements between the metallic sample and Standard Calomel Electrode (SCE) reference electrode. Moreover, some polarization steps under cathodic potential were practiced. The protocol was described in [33]. The samples were a parallelepiped of $15 \times 10 \times 10$ mm$^3$; the fretting surface was polished until reaching Sa value of $60 \pm 10$ nm. Concerning the cylindrical samples, a 1mm length was manufactured with a curvature radius of 10 mm. From a parallelepiped, $10 \times 10 \times 15$ mm$^3$, one face was machined in order to get 9 mm height on the border and keep 1mm of length at 10 mm height in the middle. This strategy was achieved in order to decrease the contact area and increase the contact pressure, i.e., maximum 500 MPa.

### 3.3. Analysis Methods

#### 3.3.1. Scanning Electron Microscope

This experimental tool was Zeiss VP55 (Zeiss, Oberkochen, Germany). In order to observe metallic samples, i.e., conductive ones, a voltage of 20 kV was used. Notwithstanding some oxide deposits, we paid attention to wrapping the metallic samples with aluminum film (the food type was sufficient). Through a standard setup these observations were efficient. However, concerning some PEEK coatings, the voltage should be decreased because of some insulating characteristics related to the PEEK material.

A voltage of 5 kV was selected. Moreover, some observations have been carried out under partial pressure in order to avoid some burning parts on the polymeric material.

### 3.3.2. 3D Profilometry

The device is named NT9100 ex. Veeco, Bruker$^{TM}$ (Bruker, San Jose, CA, USA). 3D profilometry was investigated in order to show qualitatively some wear profile. Moreover, it was the dedicated tool to obtain some quantitative data as roughness parameters and wear volume. The working surface was 1.10 μm × 0.86 μm. The wear volume was calculated by defining a reference plane without some filtering processes. Concerning the cylindrical samples, some cylindrical filters were practiced (curvature radius of 10 mm) in order to get a flat surface and the wear volume on the cylindrical shape measured in a practical way. Nonetheless, it is worth noting that, due to some localized corrosion dissolution assisted by fretting, some wear spots, i.e., wear areas, represent the total wear volume. A surface volume, complex expression, was considered. One should attempt some explanations. On every spot, the surface was calculated and the corresponding wear volume was measured. After measuring 5 spots, an average value of the surface volume was issued from 5 measurements and according to the total surface of wear, the total wear volume was reached.

### 3.3.3. SEM and Focused Ion Beam (FIB) Analysis

The SEM, Scanning Electron Microscope, was the ZEISS$^{TM}$ VP 55 (Zeiss, Oberkochen, Germany). This SEM technique was practiced to observe the wear track area. The voltage was 20 kV for observing metal. In the case related to PEEK deposit, the selected voltage was of 5 kV in order to avoid some burning effect on the polymeric material. The Focused Ion Beam (FIB) was the FIB/SEM DualBeam Thermo Scientific$^{TM}$ Helios Nanolab$^{TM}$ 600I. The ablation was managed through ion beam technique and the observation was practiced through SEM. A few experimental details are essential in order to get some key points related to the sub-surface of the wear track area. The acceleration tension was 5 kV, the current was of 0.34 nA, the magnification was around ×10,000 order of magnitude.

## 4. Coating Effect on the Tribological Behavior

With the modular junctions related to Ti-6Al-4V/Ti-6Al-4V, shown in Figure 2d, the expected coatings should be a benefit according to avoiding some fretting corrosion degradations. The fretting corrosion experiments were investigated for 16 h in bovine serum solution in order to mimic the physiological liquid. The protein concentrations respected the ISO 14242 [4], i.e., 30 g·L$^{-1}$. First of all, some interesting behavior about Ti-6Al-4V/Ti-6Al-4V contact will be presented (sometimes stick sometimes slip). Second some DLC coating effects will be described and commented according to Ti-6Al-4V/Ti-6Al-4V behavior. Third some comparisons with PEEK coatings will be investigated.

### 4.1. Ti-6Al-4V/Ti-6Al-4V, No Coating

Before studying the coating impact on Ti-6Al-4V, one paid attention on the pristine contact Ti-6Al-4V/Ti-6Al-4V. Under fretting corrosion in bovine serum solution during 16 h, the curve wear volume vs. dissipated energy provided both behavior tendencies. The Figure 4 is highlighting the total wear volume, i.e., the cylindrical sample wear volume (Ti-6Al-4V) and the flat sample wear volume (Ti-6Al-4V and Ti-6Al-4V + diamond-like carbon coating), according to the total dissipated energy. The area of the cycle tangential load vs. displacement means the dissipated energy. Every second, 1Hz of frequency, is corresponding to an amount of dissipated energy. By adding every energy through 57,600s, i.e., 16 h, the total dissipated energy was obtained. The black dots correspond to Ti-6Al-4V/Ti-6Al-4V contact. It is worth noting that the mechanical contact conditions are exactly the same between the four highlighted dots on the graph. The electrochemical conditions are quite smiliair, open circuit potential (OCP) conditions and solution composition. We are going first to focus on Ti-6Al-4V/Ti-6Al-4V contact. First of all, the physical fact that the wear volume increase is linear, according to the total dissipated energy, is highlighted in Figure 4. The tremendous evolution is related

to a drastically different trend. First, it is worth noting that A ratio is defined as dissipated energy (area of the tangential load vs. displacement / parralelogram) over the total energy (area of the rectangle max-min tangential load multiplied by min-max displacement). Both behaviors, corresponding to the lowest and the highest dissipated energies, show that behind the pseudo linear evolution, i.e., wear volume vs. total dissipated energy, a material issue is occurring.

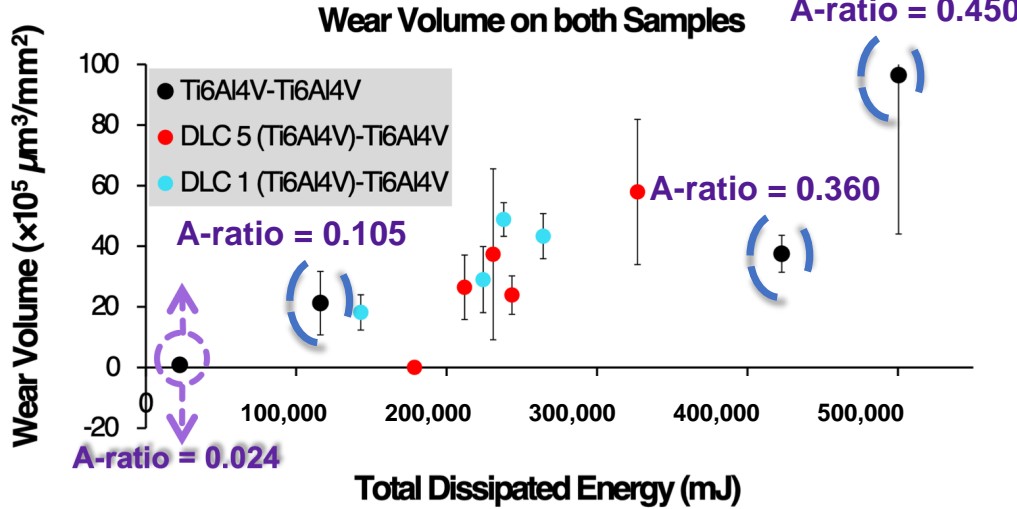

**Figure 4.** Evolution of the total wear volume ($10^5$ $\mu m^3/mm^2$), both samples in contact, according to the total dissipated energy (mJ).

The Figure 5 is showing the cross section, partially, of the flat sample corresponding to the A = 0.024 case. The Figure 5a is corresponding to a cross section corresponding to the wear track area. It is worth noting that the sub surface cracks are particularly prominent. The stick behavior enhanced some crack formations from the top surface until the maximum of the contact stresses was reached. From contact mechanics [34], the maximum contact pressure is located under the contact surface. Figure 5b highlights a crack, 2 $\mu m$ under the surface level, that dissipated the energy through a parallel orientation compared to the surface. It is more difficult to go from the top of the bulk material than to propagate in parallel with the material surface when considering a crack. About the case of A = 0.105, the stick behavior was promoted. It is worth noting that all mechanical data are consistent with the metallographical observations, Figure 5.

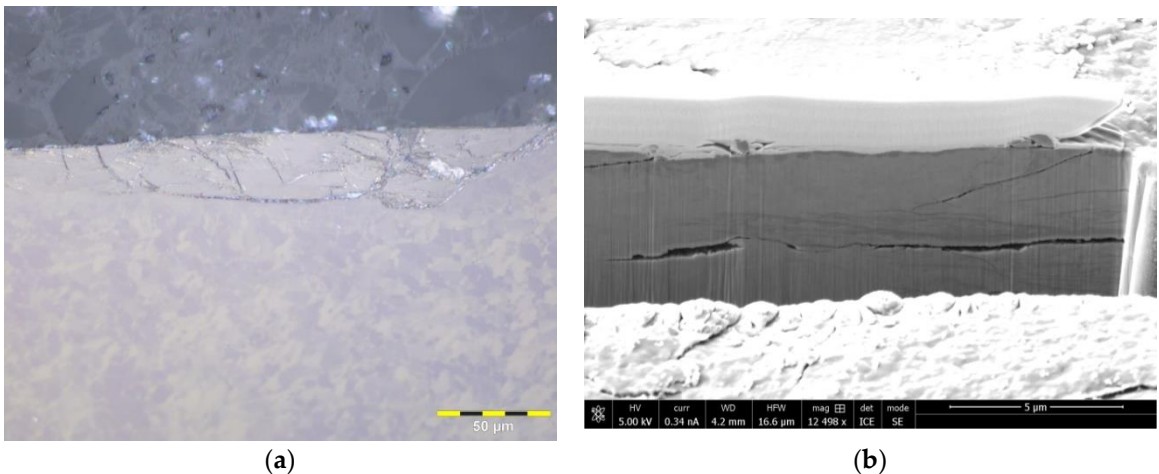

(**a**)                                                                                          (**b**)

**Figure 5.** Ti-6Al-4V: (**a**) cross section image of the wear track area of the A = 0.024 sample, smooth polishing technique; (**b**) focused ion beam (FIB) cross section of the same wear track area.

Concerning other experimental data, high wear volume-high total dissipated energy, the metallographical data are consistent with gross slip regime. A ratio is higher than 0.2. Moreover, Figure 6 is highlighting no crack on the subsurface of the material. Some wear debris are visible on the top surface without generating any crack.

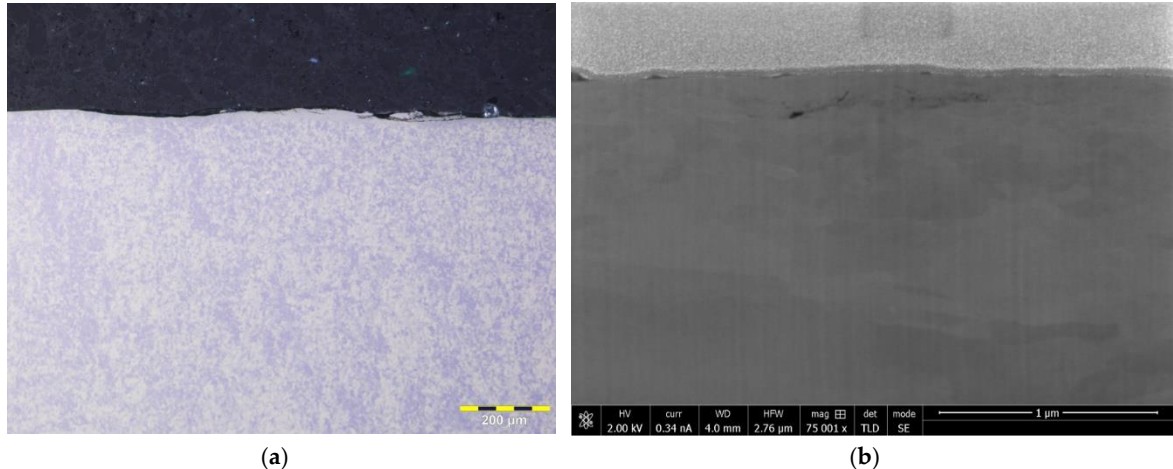

(**a**)　　　　　　　　　　　　　　　　　　　　　　　　　　　　(**b**)

**Figure 6.** Ti-6Al-4V; (**a**) cross section image of the wear track area of the A = 0.36 sample, smooth polishing technique; (**b**) FIB cross section of the same wear track area.

Figure 7 is representing the A ratio evolution vs. duration of the experiment. Higher than 0.2, it is corresponding to 0.36 and 0.45 experiments on the Figure 4, gross slip regime.

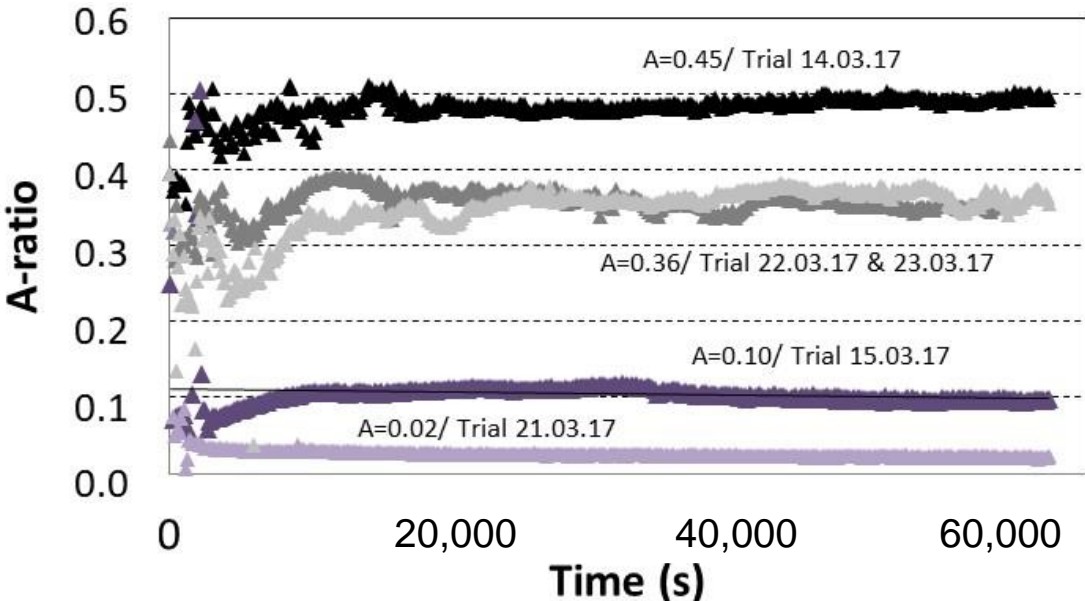

**Figure 7.** A ratio vs. time concerning Ti-6Al-4V/Ti-6Al-4V contact. A: Ed/Etot; Ed: dissipated energy, Etot: total energy, reciprocal displacement between materials in contact.

The difficult point is that every experimental (materials and mechanics) condition was exactly the same. No physical explanation, from our investigations, is available from these results. Because of practicing these experiments in an aqueous environment, one may notice that these observations are revealing the issue related to phenomena scales. Tribology, especially this kind of experiments of fretting corrosion, is a macroscopic phenomenon according to the usual experimental tool. The physical phenomena occurring during friction are related to adsorption, and adherence that produce macroscopic marks but the origins were issued from microscopic phenomena. From implants, many orthopedic

surgeons do testify that the adherence between two metallic materials made of titanium alloy, Ti-6Al-4V, occurs on the trunnion connection roughly half of the time. Sometimes the phenomenon is happening and sometimes not. Even within in vitro laboratories, this scientific phenomenon is occurring randomly. Some attempts were investigated through controlling the surface state, i.e., roughness, the impact force during assembling phase, etc. The facts delivered from orthopedic surgeons and even amongst in vitro experiments state that the same conclusion is happening: controlling the stick and the slip behavior of Ti-6Al-4V/Ti-6Al-4V may open closed doors. Through this book chapter, the authors want to describe two options, taking into account some coatings benefits, in order to control the tribology of materials.

### 4.2. Ti-6Al-4V/Ti-6Al-4V + DLC or PEEK Coating

Based on Figure 4, two DLC coatings were tested. The red and blue dots corresponded to the ongoing contact study. The first point is related to the dot with a dissipated energy lower than 20,000 mJ which has a wear volume of 0. It was not related to a measurement; it was due to the discrepancy. From the experimental protocol the wear volume measurements were not achieved by giving actual data. This matter of fact is due to adhesion. It was happening one time over five with DLC 5 coating. The wear volume vs. dissipated energy cloud is located in the same area. The results make sense and the relative straight-line growth is in accordance with other DLC coatings on Ti-6Al-4V vs. Ti-6Al-4V in dry conditions with multiple normal stresses [35–38].

The Figure 8 is highlighting the wear volume vs. the total dissipated energy concerning the flat sample, i.e., Ti-6Al-4V + coating. The coating should be DLC, i.e., diamond-like carbon, and PEEK, polyetheretherketone.

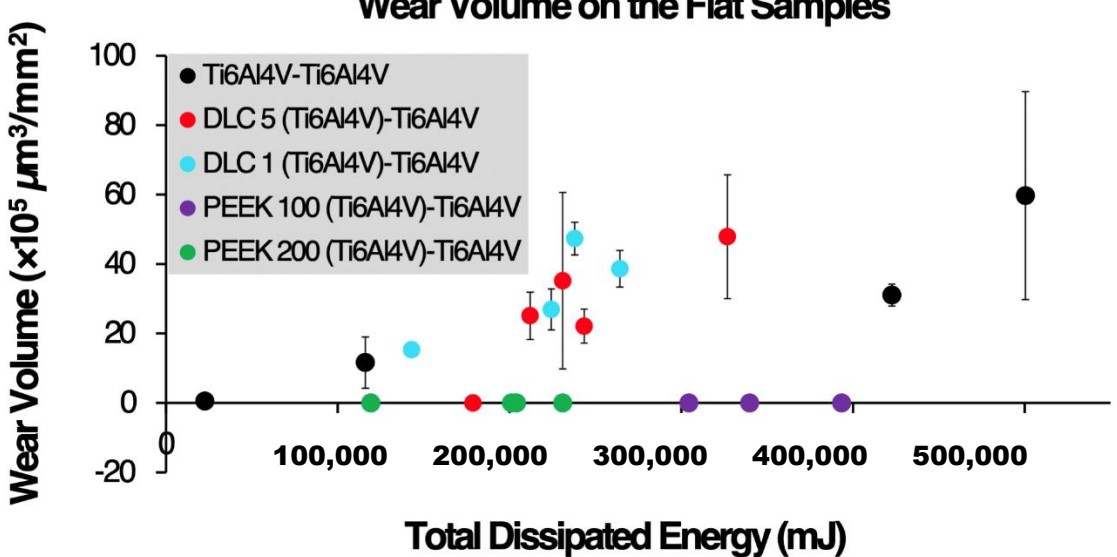

**Figure 8.** Evolution of the total wear volume ($10^5$ $\mu m^3/mm^2$), Ti-6Al-4V + coatings, according to the total dissipated energy (mJ). Polyetheretherketone (PEEK) xxx coating is representing roughly the thickness of the PEEK coating.

It is worth noting that wear volume measurement was not possible on the coating of PEEK. The wear phenomenon is not the physical phenomenon that is relevant concerning the key point of friction. Through the experimental conditions the competition between adhesion and relaxation processes is occuring, i.e., Ti-6Al-4V/PEEK contact. No experimental wear volume is measurable on PEEK coating. Inside part 3 the Schallamach waves were evoked. The difference of elastic modulii between materials in contact was between 100 and 1000 ($E_{Rubber}$ = 1–0.01 GPa and $E_{Glass}$ = 70 GPa [39]); some Schallamach waves are generated and this physical phenomenon is well described [26,40]. In our case $E_{PEEK}$ = 4 GPa and $E_{Ti\text{-}6Al\text{-}4V}$ = 115 GPa, roughly a ratio of 30 is exhibited. The ratio value between

PEEK vs. Ti-6Al-4V is weaker than the one of rubber vs. glass. Another point would concern the energy release rate from the fracture mechanics [41]. By comparing G, the energy release, the ratio between rubber vs. glass is about 1000. It means that rubber material is able to store a lot of energy before failing. On the contrary, with glass material, the brittleness of this material does exhibit that storing energy is not possible. It is worth noting that the titanium alloy against PEEK materials combination does not exhibit such ratio of energy release rate. The difference of the materials properties is not in the way to exhibit the Schallamach waves concerning the PEEK/Ti-6Al-4V contact. The values are generic values shown in Table 2.

**Table 2.** Some mechanical properties related to rubber/glass and PEEK/Ti-6Al-4V contacts. Showed values are in the right order of magnitude.

| *; ** | Rubber | Glass | PEEK | Ti-6Al-4V |
|---|---|---|---|---|
| E/ GPa | 0.01–0.1 | 70 | 4 | 115 |
| $K_{IC}$ / MPa.m$^{1/2}$ | 4 | 0.8 | 1–2 | 50 |
| $\sigma_m$ / MPa | 13 | 50 | 150 | 815–875 |
| G / J/m$^2$ | 16–100 | $9.10^{-3}$ | 1 | 25 |

* [39]; ** [42].

Notwithstanding, PEEK material is able to store some energy issued from friction phenomenon without any crack and failure that takes into account debris formation.

The Figure 9 shows the surface of PEEK 200 on Ti-6Al-4V after 16 h of fretting corrosion against Ti-6Al-4V in bovine serum. Some waviness appears on huge valleys (−15 μm) and huge mountains (+26 μm). The initial roughness was about 0.5 μm. That is the reason the authors suggest that measuring wear, i.e., issued from debris, is not feasible. No debris were highlighted after the experiment on the top surface of every material in contact.

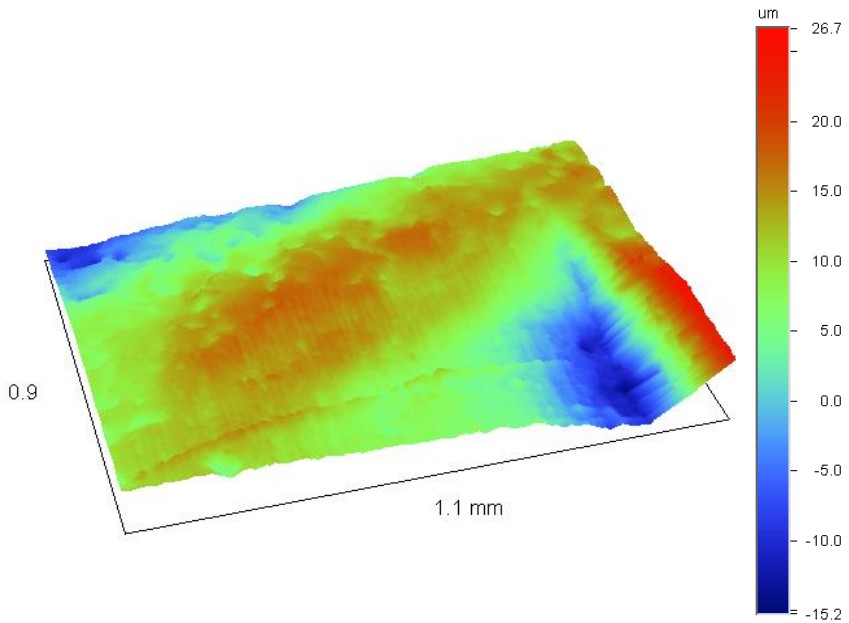

**Figure 9.** PEEK 200 on Ti-6Al-4V after 16 h of fretting corrosion in bovine serum, inside the wear track area.

The PEEK material is deformed but no wear is visible and measurable on this face and on the Ti-6Al-4V. In order to assess this experimental fact, one might have a look at the evolution of Open

Circuit Potential vs. the Standard Calomel Electrode as a reference electrode. Figure 10 is showing the OCP evolution concerning one test PEEK 200 µm on Ti-6Al-4V against Ti-6Al-4V.

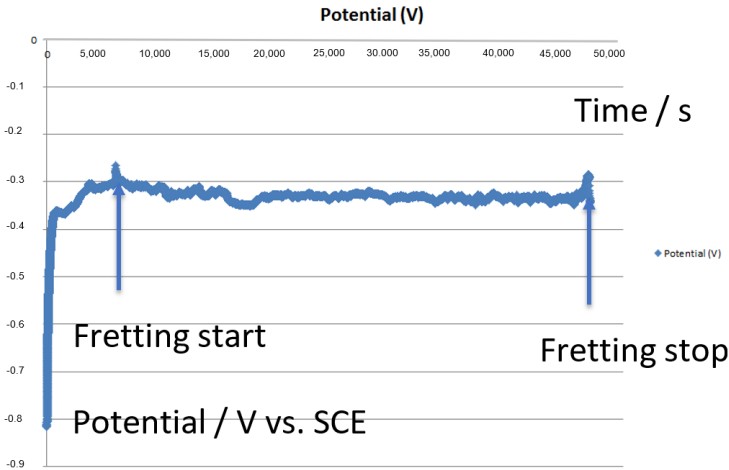

**Figure 10.** Open Circuit Potential vs. Standard Calomel Electrode (SCE) of the PEEK 200 on Ti-6Al-4V and Ti-6Al-4V assembly. A wire connection is fixed between both materials involved in the contact.

At the fretting start, it is worth noting that no OCP decrease is occuring, arrow fretting starts on the graph of Figure 10. Usually, as reported in [43,44], the OCP decreases according to the time when some friction occurs between materials solicited under friction conditions (Figure 11). Destroying the oxides layer involves 'naked' metal in contact with solution. Thus, the corrosion and dissolution of metal, was ongoing. However, from Figure 10, because the OCP is not decreasing according to the time, it means that the oxide layer is not destroyed during the fretting phase.

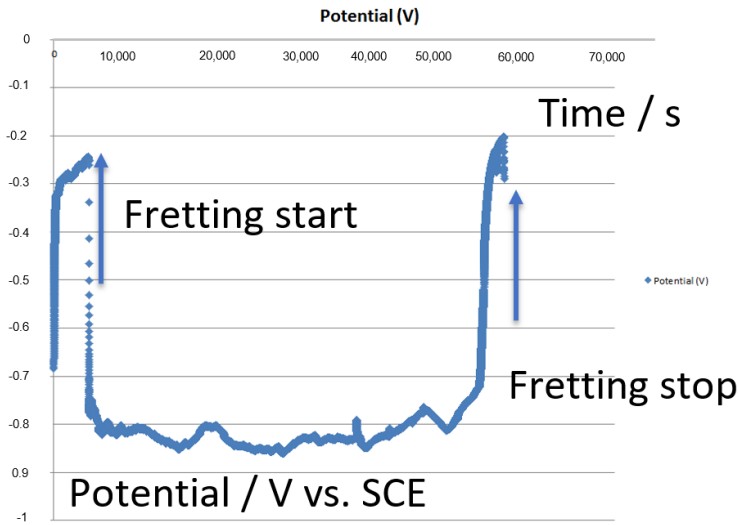

**Figure 11.** Open Circuit Potential vs. SCE of the Ti-6Al-4V and Ti-6Al-4V assembly. A wire connection is fixed between both materials involved in the contact.

The Figure 11 is highlighting the usual OCP evolution when some friction and/or fretting is occurring between metallic materials. The OCP drop is characterized as 600 mV. The major benefits of the PEEK coatings are the lack of mechanical and corrosive wear.

## 5. Conclusions and Outlooks

The ongoing study presented a new coating, a biocompatible one. The targeted field of use is the orthopedic implant, especially the hip prosthesis. Two scientific points were developed

through this book chapter. The first outline was questioning the Schallamach waves. The couple PEEK/Ti-6Al-4V does not involve the well known Schallamach waves. They are highlighted on the contact of rubber/glass. Even if there are some close marks on the PEEK surface after fretting corrosion experiments, the mechanical properties do not allow the propagation of the Schallamach waves. It is worth noting, on Figure 9, that some material deformation is highlighted after fretting corrosion tests. This material deformation did not occur with any wear. Second, no corrosive degradation was highlighted. More results on that point will be presented through next publications. The OCP did not decrease when the fretting/friction occurred between PEEK coating on Ti-6Al-4V against Ti-6Al-4V. Due to the target (the modular junction of hip prosthesis), the primary achievement should be the biocompatibility. It means no connection between metallic materials with brazing (copper, gold-silver, etc.) or any glue with epoxy resin. Through the PEEK coating in a modular junction it should be an acceptable issue in order to avoid any metal–metal junction. It needs a proof of concept in an actual case, actual modular junction, and some next investigations should be focussed on assessing the suggested solution.

**Author Contributions:** J.G.: conceptualization, funding acquisition; V.F.: funding acquisition; H.D.: investigation; K.K.: Methodology; T.T.: investigation; L.S.: investigation; S.E.-M.: investigation; P.O.: funding acquisition, supervision; J.F.: methodology, resources; P.K.: funding acquisition, supervision. All authors have read and agreed to the published version of the manuscript.

**Funding:** This research was funded by Labex Manutech Sise, grant number Mod-Hip. Some additional funding from the University of Johannesburg about Master's students grants.

**Acknowledgments:** The authors are grateful to Evonik Gmbh through Marc Knebel and M.G.P. Planche. The IREIS Company through M.L. Dubost developed a new process to improve the PEEK coating and it provides some DLC coatings. All tribological investigations were improved through the Labex Manutech Sise financial support from the Lyon University thanks to C. Donnet et al. The pristine idea related to PEEK is coming from D.D. Macdonald, Eurocorr 2009. Finally, F. Farizon and B. Boyer and their colleagues are acknowledged in order to work around the orthopedic implants field.

**Conflicts of Interest:** The authors declare no conflict of interest.

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
