# Peer review of "Some Hard or Soft Coatings to Protect the Pristine Biometallic Substrates under Fretting-Corrosion Solicitations: What Should Be the Best Solution?"

_lubricants, doi:10.3390/lubricants8050055_

Round 1

Reviewer 1 Report

I have read with a great interest your paper titled "Some hard or soft coatings to protect the pristine biometallic substrates under fretting-corrosion solicitations: what should be the best solution?". The topic is interesting and worth investigating, in my opinion, the paper in its present could be published after some modificatyion. Some suggestions are listed as following:

  1. In abstract: I.e.--->i.e.
  2. Table 1 should be rewritten.
  3. In 2.2.2 section, do not mix statement about DLC and PEEK with the Experimental section. 
  4. In 2.2.2 DLC, Diamond Like Carbon, and PEEK, PolyEtherEtherKetone.----> Diamond Like Carbon(DLC), and PolyEtherEtherKetone(PEEK). Just using DLC and PEEK afterwards.
  5. Please provide more chemical and mechanical characterizatin of DLC and PEEK coatings
  6. In line 232, "of all the physical, physical sciences, fact that" ---> of all the physical fact that
  7. Fig.7 should be replotted. Please provide the information for each line in the figure.
  8. In line 311, it is table 2
  9. Please provide more information for ref[4,5,6]

Author Response

The authors text italic

They acknowledge the reviewer for improving the text.

Reviewer 1

Comments and Suggestions for Authors

I have read with a great interest your paper titled "Some hard or soft coatings to protect the pristine biometallic substrates under fretting-corrosion solicitations: what should be the best solution?". The topic is interesting and worth investigating, in my opinion, the paper in its present could be published after some modificatyion. Some suggestions are listed as following:

  1. In abstract: I.e.--->i.e. OK

The authors are grateful about detailing some mistyping.

  1. Table 1 should be rewritten. OK

The authors are suggesting a new table/format.

  1. In 2.2.2 section, do not mix statement about DLC and PEEK with the Experimental section. 

The section material was removed through a typical part, second part.

  1. In 2.2.2 DLC, Diamond Like Carbon, and PEEK, PolyEtherEtherKetone.----> Diamond Like Carbon(DLC), and PolyEtherEtherKetone(PEEK). Just using DLC and PEEK afterwards.

Acronymous are now using

  1. Please provide more chemical and mechanical characterizatin of DLC and PEEK coatings

The reviewer’s query makes sense. Unfortunately, due to confidential agreements (ongoing at that time), some pristine and general properties were detailed, Refs [8-9] and [20-25] but some targeted chemical and mechanical properties are not allowed to be published. I hope the reviewer shall understand this statement.

  1. In line 232, "of all the physical, physical sciences, fact that" ---> of all the physical fact that

The sentence was rearranged.

  1. Fig.7 should be replotted. Please provide the information for each line in the figure.

The Figure 7 was totally rearranged with legends.

  1. In line 311, it is table 2

The numbering was checked and it was adapted.

  1. Please provide more information for ref[4,5,6

The mentioned references were detailed.

Reviewer 2 Report

It’s well known that polymer has outstanding tribology performance in some engineering applications. This study suggests a biocompatible polymer material (PEEK) to solve the material failure in the orthopedic implant, especially the hip prosthesis. The suggested solution is very interesting and should be beneficial to the next investigation on the field of the orthopedic implant. The writing is organized very well and the data are reasonable. It can be published after a minor revision. The drawbacks are according to my view:

  1. Page 5, from Line 125 to Line 135, the text is the same as the last paragraph.
  2. Page 7, in “3.1 Ti-6Al-4V…”, for the readability, please give more information about the total dissipated energy, such as define, expression…
  3. Page 7, in “3.1 Ti-6Al-4V…”, what is the wear volume in the two-material combination? Does it contain both side materials? Or it only belongs to Ti6Al4V in the tribo-pair?
  4. Page 7, in “3.1 Ti-6Al-4V…”, what is “A-ratio” in Figure 4? Three “A-ratio =” in Figure 4 and the detail values need to be shown here.
  5. Page 8, Figure 5, please point out the substrate/ DLC coating with arrows in the picture.
  6. Page 10, Line 296, the word “Trough” should be “Through”.

Author Response

The authors acknowledge the reviewer 2 in order to improve the quality of the manuscript

Authors replies in italic.

Reviewer 2

Open Review

English language and style

( ) Extensive editing of English language and style required
( ) Moderate English changes required
(x) English language and style are fine/minor spell check required
( ) I don't feel qualified to judge about the English language and style

Yes

Can be improved

Must be improved

Not applicable

Does the introduction provide sufficient background and include all relevant references?

(x)

( )

( )

( )

Is the research design appropriate?

(x)

( )

( )

( )

Are the methods adequately described?

(x)

( )

( )

( )

Are the results clearly presented?

(x)

( )

( )

( )

Are the conclusions supported by the results?

(x)

( )

( )

( )

Comments and Suggestions for Authors

It’s well known that polymer has outstanding tribology performance in some engineering applications. This study suggests a biocompatible polymer material (PEEK) to solve the material failure in the orthopedic implant, especially the hip prosthesis. The suggested solution is very interesting and should be beneficial to the next investigation on the field of the orthopedic implant. The writing is organized very well and the data are reasonable. It can be published after a minor revision. The drawbacks are according to my view:

  1. Page 5, from Line 125 to Line 135, the text is the same as the last paragraph.

The authors would like to thank the reviewer about the remark. It should be an issue related to the editing. Moreover according to the other reviewer, the paragraph was moved.

  1. Page 7, in “3.1 Ti-6Al-4V…”, for the readability, please give more information about the total dissipated energy, such as define, expression…

The authors added the definition of total dissipated energy, it was missing.

The area of the cycle tangential load vs. Displacement means the dissipated energy. Every second, 1Hz of frequency, is corresponding to an amount of dissipated energy. By adding every energy through 57600s, i.e 16 hours, the total dissipated energy was obtained.’

  1. Page 7, in “3.1 Ti-6Al-4V…”, what is the wear volume in the two-material combination? Does it contain both side materials? Or it only belongs to Ti6Al4V in the tribo-pair?

The wear volume was calculated, every test, on both samples. On Figure 4 as an example, the wear volume on both samples was considered.On Figure 8, the wear volume on the flat samples was considered, i.e. the flat surface of Ti-6Al-4V adding from nothing, DLC 1, DLC 5, PEEK 100 to PEEK 200 coating.

  1. Page 7, in “3.1 Ti-6Al-4V…”, what is “A-ratio” in Figure 4? Three “A-ratio =” in Figure 4 and the detail values need to be shown here.

It should be emphasized, the reviewer is right. Before Figure 4, A ratio was defined. The values are readable on the graph.

First, it is worth noting that A ratio is defined: dissipated energy (Area of the tangential load vs. Displacement / parralelogram) over the total energy (area of the rectangle max-min tangential load multiplied by min-max displacement).’

  1. Page 8, Figure 5, please point out the substrate/ DLC coating with arrows in the picture.

The Figure 5 is related to Ti-6Al-4V/Ti-6Al-4V, only this tribo pair, 4.1 sub part. On Figure 4, some Ti-6Al-4V+DLC/Ti-6Al-4V contcts were shown in order to compare with the Ti-6Al-4V/Ti-6Al-4V pair.

  1. Page 10, Line 296, the word “Trough” should be “Through”

Thanks, it was corrected.
